# Bounding Box Stability against Feature Dropout Reflects Detector Generalization across Environments

**Yang Yang**[1,2†], **Wenhai Wang**[3,2], **Zhe Chen**[4,2†], **Jifeng Dai**[5,2], **Liang Zheng**[1]
[1]The Australian National University    [2]OpenGVLab, Shanghai AI Laboratory
[3]The Chinese University of Hong Kong    [4]Nanjing University    [5]Tsinghua University
{yang.yang3@anu.edu.au,whwang@ie.cuhk.edu.hk,czcz94cz@gmail.com}
{daijifeng@tsinghua.edu.cn,liang.zheng@anu.edu.au}

## Abstract

Bounding boxes uniquely characterize object detection, where a good detector gives accurate bounding boxes of categories of interest. However, in the real-world where test ground truths are not provided, it is non-trivial to find out whether bounding boxes are accurate, thus preventing us from assessing the detector generalization ability. In this work, we find under feature map dropout, good detectors tend to output bounding boxes whose locations do not change much, while bounding boxes of poor detectors will undergo noticeable position changes. We compute the box stability score (BoS score) to reflect this stability. Specifically, given an image, we compute a normal set of bounding boxes and a second set after feature map dropout. To obtain BoS score, we use bipartite matching to find the corresponding boxes between the two sets and compute the average Intersection over Union (IoU) across the entire test set. We contribute to finding that BoS score has a strong, positive correlation with detection accuracy measured by mean average precision (mAP) under various test environments. This relationship allows us to predict the accuracy of detectors on various real-world test sets without accessing test ground truths, verified on canonical detection tasks such as vehicle detection and pedestrian detection. Code and data are available at https://github.com/YangYangGirl/BoS.

## 1 Introduction

Object detection systems use bounding boxes to locate objects within an image (Ren et al., 2015; Lin et al., 2017; He et al., 2017), and the quality of bounding boxes offers an intuitive understanding of how well objects of a class of interest, e.g., vehicles, are detected in an image. To measure detection performance, object detection tasks typically use the mean Average Precision (mAP) (Everingham et al., 2009) computed between predicted and ground-truth bounding boxes. *However, it is challenging to determine the accuracy of bounding boxes in the absence of test ground truths in real-world scenarios, which makes it difficult to evaluate the detector generalization ability.*

In this work, we start with a specific aspect of bounding box quality, i.e., bounding box stability under feature perturbation. Specifically, given an image and a trained detector, we apply Monte Carlo (MC) dropout (Gal & Ghahramani, 2016) in the testing processing, which randomly set some elements in certain backbone feature maps to zero. Afterward, we use bipartite matching to find the mapping of bounding boxes with and without dropout perturbations, and then compute their intersection over union (IoU), called box stability score (BoS score). As shown in Fig. 1(a) and (b), under feature perturbation, bounding boxes remaining stable (with high BoS score) would indicate a relatively good prediction, while bounding boxes changing significantly (with low BoS score) may suggest poor predictions.

Based on this idea, we contribute to discovering *strong correlation between bounding box stability and detection accuracy under various real-world test environments*. Specifically, given a trained de-

---

†This work is done when they are interns at Shanghai AI Laboratory.

tector and a set of synthesized sample sets, we compute BoS score and actual mAP over each sample set. As the strong correlation (i.e., coefficients of determination $R^2 > 0.94$) shown in Fig. 1(c), a detector overall would yield highly stable bounding boxes and thus have good accuracy (mAP) in relative environments, and in difficult environments where the detector exhibits low bounding box stability, object detection performance can be poor.

This finding endows us with the capacity to estimate detection accuracy in various test environments without the need to access ground truth. This is because the BoS score (1) does not rely on labels in computation, which encodes the difference between original predictions and predictions after feature dropout, and (2) is strongly correlated with detection accuracy. In fact, it is well known that detection accuracy drops when the test set has different distribution from the training set. That said, without ground truths, it is challenging how to estimate test accuracy under distribution shifts. As a use case, being able to obtain model accuracy without ground truths allows us to efficiently assess systems deployed in new environments.

Several methods (Deng & Zheng, 2021; Guillory et al., 2021; Deng et al., 2021; Sun et al., 2021; Li et al., 2021; Garg et al., 2022), collectively referred to as AutoEval, have emerged to predict the classifier performance in new unlabeled environments. But these methods are poor at predicting detector performance because they do not take into account the regression characterization of the detection task. With the aforementioned box stability score measurement, we can effectively estimate the detection accuracy of a detector without ground truth. It is also the first method for AutoEval in object detection.

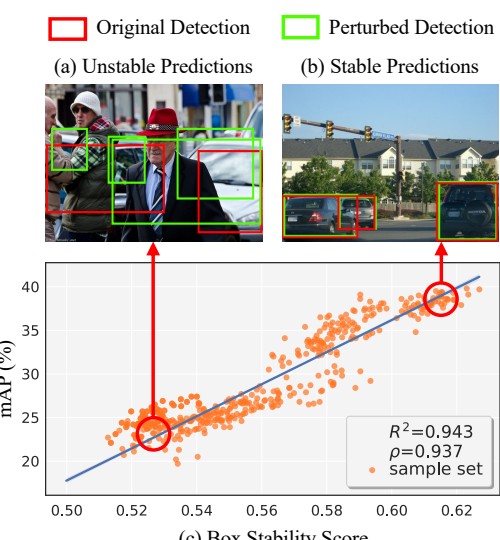

Figure 1: Visual examples illustrating the correlation between bounding box stability and detection accuracy. (a) Bounding boxes change significantly upon MC dropout, where we observe complex scenarios and incorrect detection results. (b) Bounding boxes remain at similar locations after dropout, and relatively correct detection results are observed. (c) We observed that the box stability score is positively correlated to mAP (coefficients of determination $R^2 > 0.94$, Spearman's Rank Correlation $\rho > 0.93$).

In sum, our main contributions are as follows:

(1) We report a strong positive correlation between the detection mAP and BoS score. Our proposed BoS score calculates the stability degree for the bounding box with and without feature perturbation by bipartite matching, which is an unsupervised and concise measurement.

(2) We demonstrate an interesting use case of this strong correlation: estimating detection accuracy in terms of mAP without test ground truths. To our best knowledge, we are the first to propose the problem of unsupervised evaluation of object detection. For the first time, we explore the feasibility of applying existing AutoEval algorithms for classification to object detection.

(3) To show the effectiveness of the BoS score in estimating detection mAP, this paper conducts experiments on two detection tasks: vehicle detection and pedestrian detection. For both tasks, we collate existing datasets that have the category of interest and use leave-one-out cross-validation for mAP estimator building and test mAP estimation. On these setups, we show that our method yields very competitive mAP estimates of four different detectors on various unseen test sets. BoS score achieves only 2.25% RMSE on mAP estimation of the vehicle detection, outperforming confidence-based measurements by at least 2.21 points.

## 2 RELATED WORK

**Mainstream object detectors** are often categorized into two types, two-stage detectors (Ren et al., 2015; Cai & Vasconcelos, 2019) predict boxes w.r.t. proposals, and single-stage methods make pre-

dictions w.r.t. anchors (Lin et al., 2017) or a grid of possible object centers (Tian et al., 2019; Duan et al., 2019). Recently, transformer-based methods have shown an incredible capacity for object detection, including DETR (Carion et al., 2020), Deformable DETR (Zhu et al., 2020), and DINO (Zhang et al., 2022), etc. In the community, numerous popular datasets are available for benchmarking detector performance, such as COCO (Lin et al., 2014) and ADE20K (Zhou et al., 2019), and so on. Nonetheless, due to the distribution shifts, the results on test sets in public benchmarks may not accurately reflect the performance of the model in unlabeled real-world scenarios. To address this issue, we focus on studying detector generalization ability on test sets from various distributions.

**Label-free model evaluation** predicts performance of a certain model on different test sets (Madani et al., 2004; Donmez et al., 2010; Platanios et al., 2016; 2017; Jiang et al., 2019). Some works (Guillory et al., 2021; Saito et al., 2019; Hendrycks & Gimpel, 2016) indicate classification accuracy by designing confidence-based measurements. In the absence of image labels, recent works (Deng & Zheng, 2021; Deng et al., 2021; Sun et al., 2021) estimated model performance for AutoEval with regression. Deng & Zheng (2021); Sun et al. (2021); Peng et al. (2024) used feature statistics to represent the distribution of a sample dataset to predict model performance. Deng et al. (2021) estimated classifier performance from the accuracy of rotation prediction. ATC (Garg et al., 2022) predicted accuracy as the fraction of unlabeled examples for which the model confidence exceeds the learned threshold. Different from the above works designed for the classification task, we aim to study a brand-new problem of estimating the detector performance in an unlabeled test domain.

**Model generalization** aims to estimate the generalization error of a model on unseen images. Several works (Neyshabur et al., 2017; Jiang et al., 2018; Dziugaite et al., 2020) proposed various complexity measures on training sets and model parameters, to predict the performance gap of a model between a given pair of training and testing sets. Neyshabur et al. (2017) derived bounds on the generalization gap based on the norms of the weights across layers. Jiang et al. (2018) designed a predictor for the generalization gap based on layer-wise margin distributions. Corneanu et al. (2020) derived a set of topological summaries to measure persistent topological maps of the behavior of deep neural networks, and then computed the testing error without any testing dataset. Notably, these methods assume that the training and testing distributions are identical. In contrast, our work focuses on predicting the model performance on test sets from unseen distributions.

## 3 STRONG CORRELATION BETWEEN BOUNDING BOX STABILITY AND DETECTION ACCURACY

### 3.1 MEAN AVERAGE PRECISION: A BRIEF REVISIT

Mean average precision (mAP) (Everingham et al., 2009) is widely used in object detection evaluation. We first define a labeled test set $\mathcal{D}^l = \{(x_i, y_i)\}_{i=1}^{M}$, in which $x_i$ is an image, $y_i$ is its ground truth label, and $M$ is the number of images. Given a trained detector $f_\theta : x_i \to \tilde{y_i}$, which is parameterized by $\theta$ and maps an image $x_i$ to its predicted objects $\tilde{y_i}$. Since the $\mathcal{D}^l$ is available, we can obtain the precision and recall curves of the detector by comparing the predicted objects $\tilde{y_i}$ with the ground truths $y_i$. The Average Precision (AP) summarises the area under the precision/recall curve. $mAP_{50}$ and $mAP_{75}$ correspond to the average AP over all $K$ categories under 0.50, 0.75 IoU threshold, respectively. mAP corresponds to the average AP for preconfigured IoU thresholds.

In unsupervised detector evaluation, given a detector $f_\theta$ and an unlabeled dataset $\mathcal{D}^u = \{x_i\}_{i=1}^{M}$, we want to find a mAP predictor $\mathcal{A} : (f_\theta, \mathcal{D}^u) \to \text{mAP}$, which can output an estimated $\text{mAP} \in [0, 1]$ of detector $f_\theta$ on this test set:

$$\text{mAP}_{\text{unsup}} = \mathcal{A}(f_{\boldsymbol{\theta}}, \mathcal{D}^u). \tag{1}$$

### 3.2 MEASUREMENT OF BOUNDING BOX STABILITY

This paper uses an existing technique (MC dropout (Gal & Ghahramani, 2016)) for measurement. Let us define $N_{\text{ori}}$ as the number of objects detected in an image by the original detector, $N_{\text{per}}$ as the number of objects detected in the same image by the perturbed detector with MC dropout. Let $N$ and $N_{\text{max}}$ represent the smaller or larger value between $N_{\text{ori}}$ and $N_{\text{per}}$, respectively. On this basis, we denote the set with the fewest objects in the predicted set as $y = \{y_j\}_{j=1}^{N}$, and the most as

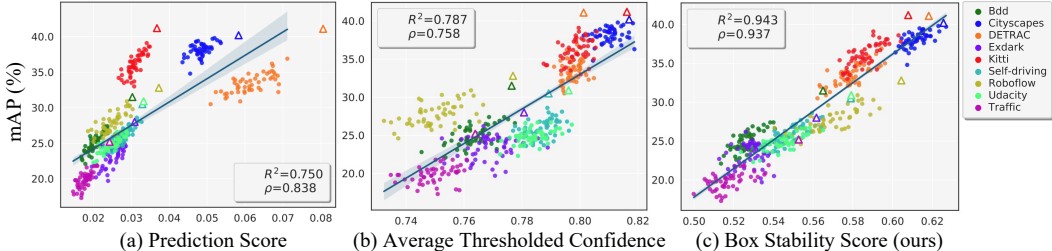

Figure 2: Correlation between different measurements and mAP. "△" of different colors represents nine real-world datasets, such as BDD (Yu et al., 2020) and Cityscapes (Cordts et al., 2016), which are used as seeds to generate sample sets by various transformations. Each point "●" of different colors represents a sample set generated from different seed sets. mAP is obtained by RetinaNet (Lin et al., 2017) trained on the COCO training set. Three measurements including two existing methods (Hendrycks & Gimpel, 2016; Garg et al., 2022) and our box stability score are used in (a), (b), and (c). The trend lines are computed by linear regression, from which we can observe a strong linear relationship (coefficients of determination $R^2 > 0.94$, Spearman's Rank Correlation $\rho > 0.93$) between box stability score and mAP.

$\hat{y} = \{\hat{y}_j\}_{j=1}^{N_{\max}}$. To find a bipartite matching between these two sets, we search for a permutation of $N$ elements $\sigma \in \mathfrak{S}_N$ with the lowest cost:

$$\hat{\sigma} = \arg\min_{\sigma \in \mathfrak{S}_N} \sum_j^N \mathcal{L}_{\text{match}}\left(y_j, \hat{y}_{\sigma(j)}\right), \tag{2}$$

where $\mathcal{L}_{\text{match}}\left(y_j, \hat{y}_{\sigma(j)}\right)$ is a pair-wise matching cost between the original prediction and the perturbed prediction with index $j$ and index $\sigma(j)$, respectively. This optimal assignment is computed efficiently with the Hungarian algorithm following previous works (Kuhn, 1955; Carion et al., 2020).

The matching cost considers the similarity between the predicted boxes of the original detector and the perturbed detector. However, using the widely-used $l_1$ loss will lead to different scales for small and large boxes, even if their relative errors are similar. Therefore, we adopt the GIoU loss (Rezatofighi et al., 2019) and define the matching loss $\mathcal{L}(y, \hat{y})$ as:

$$\mathcal{L}(y, \hat{y}) = \frac{\sum_j^N \mathcal{L}_{\text{giou}}(b_j, \hat{b}_{\sigma_{(j)}})}{N}, \tag{3}$$

where $b_j$ and $\hat{b}_{\sigma_{(j)}} \in [0, 1]^4$ denote a pair of matched boxes with index $j$ and index $\sigma_{(j)}$, respectively. After that, we average the matching loss over all $M$ test images in $\mathcal{D}^u$ and obtain the box stability score of the unseen test set $\mathcal{D}^u$:

$$\text{BS}(\mathcal{D}^u) = \frac{\sum_{i=1}^M \mathcal{L}_i}{M}, \tag{4}$$

## 4 USE CASE: LABEL-FREE DETECTOR EVALUATION

Following the practice in (Deng & Zheng, 2021), we formulate label-free evaluation as a regression problem. First, we create a training meta-dataset composed of fully labeled sample sets which are transformed versions of seed sets. Second, for each sample set, we compute the mAP and BoS score and train a regression model using BoS score as input and output mAP. Third, we use real-world test sets which are unseen during training, for mAP estimation.

### 4.1 TRAINING META-DATASET

Take mAP estimation on vehicle detection as an example. We generate 500 sample sets from 10 seed sets as mentioned in Section 5.4. We employ the leave-one-out evaluation procedure, where each time we use 450 labeled sample sets $\{\mathcal{D}_1^l, \mathcal{D}_2^l, ...\}$ generated from 9 out of 10 sources as the training meta-set. All images from the remaining 1 source serve as a real-world test set $\mathcal{D}^u$. Therefore, there are 10 different permutations to construct the training meta-set and test set. We set the maximum number of testing images to 5,000, as the number of images in the test set differs in each permutation.

## 4.2 REGRESSION MODEL TRAINING AND REAL-WORLD TESTING

From the strong correlation of box stability score and mAP shown in Fig. 2(c), we train an mAP estimator (i.e., linear regression model) on the previously constructed training meta-set to fit this relationship. Given detector $f_\theta$, the linear regression model $\mathcal{A}_{\text{linear}}$ uses its BoS score as input and the corresponding mAP as target. The $i$-th training sample of $\mathcal{A}_{\text{linear}}$ can be denoted as $\{\text{BS}_i, \text{mAP}_i\}$. The linear regression model $\mathcal{A}_{\text{linear}}$ is written as,

$$\text{mAP}_i = \mathcal{A}_{\text{linear}}(f_\theta, f_\theta', \mathcal{D}_i^l) = \omega_1 \text{BS}_i + \omega_0, \tag{5}$$

where $f_\theta'$ denotes the perturbed detector from $f_\theta$, and $\omega_0, \omega_1$ are parameters of the linear regression model. We use a standard least square loss to optimize this regression model. During real-world testing, given an unlabelled test set $\mathcal{D}^u$ and detector $f_\theta$, we compute the BoS score and then predict detector accuracy $\widetilde{\text{mAP}}$ using the trained mAP estimator.

## 5 EXPERIMENT

### 5.1 EXPERIMENTAL DETAILS

**Datasets.** We study label-free detector evaluation on two canonical object detection tasks in the main experiment: vehicle detection and pedestrian detection, to verify our method. For vehicle detection, we use 10 datasets, including COCO (Lin et al., 2014), BDD (Yu et al., 2020), Cityscapes (Cordts et al., 2015), DETRAC (Wen et al., 2020), Exdark (Loh & Chan, 2019) Kitti (Geiger et al., 2013), Self-driving (Kaggle, 2020a), Roboflow (Kaggle, 2022), Udacity (Kaggle, 2021), Traffic (Kaggle, 2020b). For pedestrian detection, we use 9 datasets, including COCO (Lin et al., 2014), Caltech (Griffin et al., 2007), Crowdhuamn (Shao et al., 2018), Cityscapes (Cordts et al., 2015), Self-driving (Kaggle, 2020a), Exdark (Loh & Chan, 2019), EuroCity (Braun et al., 2019), Kitti (Geiger et al., 2013), CityPersons (Zhang et al., 2017). Note that the annotation standard of the datasets on the category of interest should be uniform.

**Metrics.** Following (Deng & Zheng, 2021), we use root mean squared error (RMSE) as the metric to compute the performance of label-free detector evaluation. RMSE measures the average squared difference between the estimated mAP and ground truth mAP. A small RMSE corresponds to good performance. Our main experiments are repeated 10 times due to the stochastic nature of dropout, after which mean and standard deviation are reported.

**Detectors.** In the main experiment, we employ pre-trained ResNet-50 (He et al., 2016) as the backbone and one-stage RetinaNet (Lin et al., 2017) as the detection head. Some other detector architectures, including Faster R-CNN (Ren et al., 2015) with R50, Faster R-CNN with Swin (Liu et al., 2021), RetinaNet with Swin, are also compared in Table 4 (left).

**Compared methods.** Existing methods in label-free model evaluation, including prediction score (PS) (Hendrycks & Gimpel, 2016), entropy score (ES) (Saito et al., 2019), average confidence (AC) (Guillory et al., 2021), ATC (Garg et al., 2022), and Fréchet distance (FD) (Deng & Zheng, 2021), are typically experimented on image classification. For comparison, we extend them to object detection. Because PS, ES, AC, and ATC are based on the softmax output, we use the softmax output of the detected bounding boxes to compute these scores. To compute FD, we use the feature from the last stage of the backbone to calculate the train-test domain gap. Hyperparameters of all the compared methods are selected on the training set, such that the training loss is minimal.

**Variant study: comparing bounding box stability with confidence stability.** Using the same MC dropout and bipartite matching strategy, we can also measure the changes in bounding box softmax output instead of bounding box positions. In Fig. 3(c), we compare this variant with BoS score of their capability in estimating detection accuracy. Results show that the bounding box stability score is superior to confidence stability score. A probable reason is that confidence stability score does not change much due to over-confidence even if the bounding box changes significantly. So it is less indicative of detection performance.

| Method | COCO 34.2 | BDD 31.5 | Cityscapes 40.2 | DETRAC 41.1 | Exdark 28.0 | Kitti 41.2 | Self-driving 30.5 | Roboflow 32.8 | Udacity 30.9 | Traffic 25.2 | Avg. RMSE ↓ |
|---|---|---|---|---|---|---|---|---|---|---|---|
| PS (Hendrycks & Gimpel, 2016) | 8.14 | 3.22 | 5.50 | 15.83 | 0.81 | 11.80 | 1.26 | 2.48 | 1.54 | 1.99 | 5.26 |
| ES (Saito et al., 2019) | 7.06 | **1.37** | 8.40 | 16.20 | **0.14** | 13.08 | 0.93 | 3.14 | **1.23** | 3.60 | 5.52 |
| AC (Guillory et al., 2021) | 6.67 | 3.32 | 9.60 | 30.95 | 1.26 | 13.93 | 2.31 | **1.77** | 2.92 | 3.30 | 7.60 |
| ATC (Garg et al., 2022) | 10.35 | 3.20 | 5.83 | 8.21 | 1.44 | 6.34 | **0.91** | 5.13 | 1.94 | **1.23** | 4.46 |
| FD (Deng & Zheng, 2021) | 9.17 | 2.78 | 13.03 | 12.29 | 5.94 | 14.80 | 2.22 | 4.48 | 1.32 | 4.14 | 7.02 |
| BoS (ours) | **1.26**±0.32 | 1.90±0.06 | **0.89**±0.28 | 1.84±0.05 | 1.47±0.19 | **4.38**±0.07 | 1.92±0.09 | 4.79±0.10 | 1.61±0.06 | 2.43±0.09 | 2.25±0.13 |
| BoS + PS | 2.11±0.29 | 2.03±0.06 | 1.13±0.26 | **0.04**±0.02 | 1.42±0.18 | 5.56±0.06 | 1.71±0.08 | 4.69±0.10 | 1.42±0.05 | 2.24±0.09 | **2.24**±0.12 |

Table 1: Method comparison of mAP estimation for vehicle detection. Specifically, we collect 10 source datasets, including COCO (Lin et al., 2014), BDD (Yu et al., 2020), and so on. For each column, we keep one original dataset as the unseen test set, and synthesize the meta-set using remained 9 sources. Given a detector trained on a COCO training set, we report its ground truth mAP (%) on different unseen test sets in the header and employ RMSE (%) to measure the accuracy of the mAP estimation. For "Prediction Score" and "Entropy Score" methods, boxes with high prediction scores or low entropy scores are regarded as being detected correctly, respectively.

| Method | COCO 26.7 | Caltech 16.2 | CrowdHuman 33.5 | Cityscapes 19.0 | Self-driving 16.4 | Exdark 21.6 | EuroCity 16.8 | Kitti 13.3 | CityPersons 11.8 | Avg. RMSE ↓ |
|---|---|---|---|---|---|---|---|---|---|---|
| PS (Hendrycks & Gimpel, 2016) | 8.49 | 3.68 | 6.55 | 0.76 | 3.20 | 5.58 | 0.86 | **0.99** | 6.28 | 4.04 |
| ES (Saito et al., 2019) | 5.91 | **1.02** | 9.44 | 1.06 | 1.63 | 3.64 | **0.13** | 2.23 | 5.71 | 3.42 |
| AC or ATC (Guillory et al., 2021) | 5.25 | 1.38 | 17.25 | 0.79 | **1.33** | 3.18 | 0.39 | 1.69 | 5.99 | 4.13 |
| FD (Deng & Zheng, 2021) | 3.88 | 3.62 | **3.29** | 5.19 | 3.41 | 3.37 | 3.93 | 2.98 | **4.21** | 3.76 |
| BoS (ours) | **2.26**±0.05 | 2.53±0.07 | 6.39±0.02 | **0.18**±0.11 | 4.42±0.06 | **0.74**±0.07 | 1.72±0.04 | 5.16±0.08 | 6.26±0.15 | **3.29**±0.07 |
| BoS + PS | 7.04±0.05 | 2.18±0.07 | 4.63±0.02 | 0.43±0.11 | 3.29±0.06 | 0.58±0.07 | 2.33±0.04 | 3.99±0.08 | 6.88±0.15 | 3.48±0.07 |

Table 2: Method comparison of mAP estimation for pedestrian detection. The settings are same as those in Table 1, except that datasets are collected from COCO (Lin et al., 2014), Caltech (Griffin et al., 2007), Crowdhuman (Shao et al., 2018), CityPersons (Zhang et al., 2017) et al.9 datasets and the detector is trained on a CrowdHuman training set.

## 5.2 MAIN EVALUATION

**Comparison with state-of-the-art methods.** We compare our method with existing accuracy evaluators that are adapted to the detection problem. After searching optimal thresholds on the training meta-set, we set $\tau_1 = 0.4$, 0.95, and 0.3 for ATC (Garg et al., 2022), PS (Hendrycks & Gimpel, 2016), and ES (Saito et al., 2019), respectively. As mAP estimation for vehicle detection is shown in Table 1, through the leave-one-out test procedure, our method achieves the lowest average RMSE = 2.25% over ten runs. The second-best mAP estimator, ATC, yields an average RMSE = 4.46%. Across the ten test sets, our method gives consistently good mAP estimations, evidenced by the relatively low standard deviation of 0.13%. Specifically, on COCO, BDD, and Cityscapes, our method yields the lowest RMSE of 1.26%, 1.90%, 0.89%, respectively.

In Table 2, our method is compared with existing ones on label-free evaluation of pedestrian detection. We have similar observations, where BoS score makes consistent predictions across the six test scenarios, leading to the lowest average RMSE. These results demonstrate the usefulness of our key contribution: the strong correlation between BoS score and mAP.

**Using BoS score for multi-class detectors**. BoS score can be easily extended to multi-class detectors theoretically because of its class-wise nature. To verify this, we should collect *many* object

| Method | COCO | | Cityscapes | | Exdark | | Kitti | | Self-driving | | Avg. RMSE ↓ |
|---|---|---|---|---|---|---|---|---|---|---|---|
| | person 43.08 | car 33.98 | person 23.88 | car 39.66 | person 23.94 | car 28.05 | person 11.85 | car 43.66 | person 14.68 | car 31.66 | |
| PS | 5.33 | 2.55 | 1.49 | 1.07 | 6.49 | 3.92 | 2.26 | 17.47 | 5.94 | 0.37 | 4.69 |
| ES | 15.15 | 2.21 | 2.35 | 1.21 | **0.11** | 2.47 | 4.71 | 16.74 | **1.69** | **0.04** | 4.67 |
| AC or ATC | 11.58 | 3.74 | 1.86 | 1.68 | 3.70 | 3.46 | 3.20 | 16.72 | 3.97 | 1.38 | 5.13 |
| FD | 10.50 | 3.79 | 1.93 | **0.90** | 5.71 | 2.84 | 3.35 | 13.55 | 6.79 | 2.91 | 5.23 |
| BoS (ours) | **5.30**± 0.58 | **1.75**± 0.80 | **1.45**± 0.90 | 7.72± 0.51 | 1.88± 1.33 | 1.65± 0.72 | 4.55± 0.53 | 12.53± 0.67 | 2.68± 0.61 | 3.41± 0.38 | **4.29**± 0.70 |
| BoS + PS | 11.16± 0.34 | 2.57± 0.90 | 2.04± 0.71 | 5.46± 0.53 | 7.09± 1.44 | **1.13**± 0.68 | **2.88**± 0.20 | **10.53**± 0.73 | 2.26± 0.47 | 3.91± 0.57 | 4.90± 0.66 |

Table 3: Method comparison of mAP estimation for multi-class detection. The settings are same as those in Table 1, except we implemented BoS score on the 5 datasets that contain pedestrians and vehicles simultaneously, including COCO, Cityscapes, Exdark and so on.

| Method | RetinaNet | | Faster R-CNN | |
| --- | --- | --- | --- | --- |
| | Swin-T | ResNet-50 | Swin-T | ResNet-50 |
| PS | 5.71 | 5.26 | 5.34 | 5.21 |
| ES | 5.26 | 6.26 | 6.63 | 5.84 |
| AC | 6.27 | 7.60 | 5.61 | 5.02 |
| ATC | **3.44** | 4.46 | 5.36 | 5.85 |
| FD | 8.70 | 7.02 | 8.54 | 7.44 |
| BoS (ours) | 4.69±0.15 | **2.25**±0.13 | **4.83**±0.28 | **3.79**±0.12 |

| Method | RetinaNet + ResNet-50 | | |
| --- | --- | --- | --- |
| | mAP | $mAP_{50}$ | $mAP_{75}$ |
| PS | 5.26 | 10.13 | 5.36 |
| ES | 6.25 | 10.33 | 6.01 |
| AC | 7.60 | 12.86 | 7.89 |
| ATC | 4.46 | 6.36 | 6.92 |
| FD | 7.02 | 10.87 | 8.84 |
| BoS (ours) | **2.25**±0.13 | **4.84**±0.18 | **3.47**±0.15 |

Table 4: Left: mAP prediction comparison under four detector structures. All the detectors are trained on COCO. Row 1: detector heads, *i.e.*, RetinaNet and Faster R-CNN. Row 2: backbones, *i.e.*, Swin-T and ResNet-50. Right: Comparing methods of their capabilities in predicting mAP, $mAP_{50}$ and $mAP_{75}$. We use the retinanet+R50 detector trained on COCO. RMSE (%) is reported.

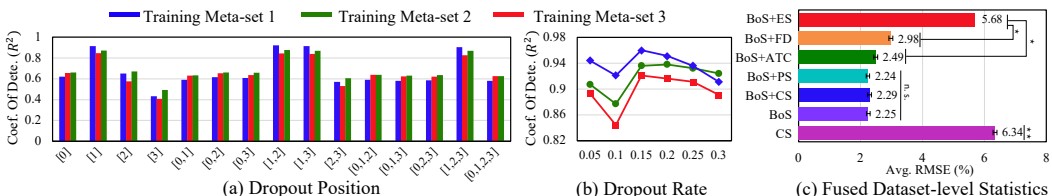

(a) Dropout Position   (b) Dropout Rate   (c) Fused Dataset-level Statistics

Figure 3: Comparing different (a) dropout positions and (b) dropout rate. We plot the coefficients of determination $R^2$ obtained during training under different dropout configurations and matching times, measured on three different training meta-sets. We use greedy search for their optimal values during training, so that $R^2$ is maximized. "[0]" means adding a dropout layer after stage 0 of the backbone, "[0, 1]" means adding a dropout after stage 0 and stage 1 of the backbone respectively, and so on. (c) AutoEval performance (RMSE, %) of confidence stability score (CS score), box stability score (BoS score), and fused dataset-level statistics. No obvious improvement is observed after fusion. "n.s." represents that the difference between results is not statistically significant (i.e., $p - \text{value} > 0.05$). $\star$ corresponds to statistically significant (i.e., $0.01 < p - \text{value} < 0.05$). $\star\star$ means the difference between results is statistically very significant (i.e., $0.001 < p - \text{value} < 0.01$).

detection datasets that have *diverse appearances* and the *same label space*. We tried and found 5 datasets that allow us to do pedestrian and vehicle detection. We implement BoS and report results in Table 3, where BoS is shown to be competitive compared with existing methods. Besides, we observe that all methods might not work as well in multi-category as it does in single-class in some datasets. This is because when confronted with multi-class mAP predictions, the features between different categories may interfere with each other.

**Effectiveness of our method for different detector structures.** Apart from the detector structure composed of RetinaNet (Lin et al., 2017) head with ResNet-50 (He et al., 2016) backbone evaluated in Table 1, we further apply this method for other structures such as Faster R-CNN (Ren et al., 2015) head with Swin (Liu et al., 2021) backbone on vehicle detection, and results are summarized in Table 4 (left). Compared with existing methods, our method yields very competitive mAP predictions. For example, using ResNet-50 backbone and RetinaNet as head, the mean RMSE of our method is 2.25%, while the second best method ATC gives 4.46%. Our method is less superior under the Swin-T backbone. It is perhaps because the transformer backbone is different from the convolutional neural networks *w.r.t* attention block, which may not be susceptible to dropout perturbation.

**Effectiveness of combining BoS score with existing dataset-level statistics.** To study whether existing label-free evaluation methods are complementary, we combine each of Entropy Score, Prediction Score, ATC, and FD with the proposed BoS score. As shown in Fig. 3(c), combining existing statistics with BoS score does not result in a noticeable improvement. For example, the average RMSE of the "box stability score + prediction score" is on the same level as using BoS score alone. It suggests that confidence in this form does not add further value to bounding box stability, which is somehow consistent with the findings in the previous paragraph. Performance even deteriorates when we use other measurements such as Entropy, FD, and ATC. While current measures based on confidence are not complementary, we believe it is an intriguing direction for further study.

**Estimating $mAP_{50}$ and $mAP_{75}$ without ground truths**. This paper mainly predicts the mAP score, which averages over 10 IoU thresholds, i.e., 0.50, 0.55, ..., 0.95. Here, we further predict $mAP_{50}$ and $mAP_{75}$, which use a certain IoU threshold of 0.50 and 0.75, respectively. In Table 4 (right), we observe that BoS score is also very useful in predicting $mAP_{50}$ and $mAP_{75}$, yielding the mean RMSE of 4.84% and 3.47%, respectively, which are consistently lower than the compared methods.

### 5.3 FURTHER ANALYSIS

**Impact of the stochastic nature of dropout.** Because dropout disables some elements in the feature maps randomly, in main experiments such as Table 1, Table 2 and Table 4, we use multiple runs of dropout, train multiple regressors and report the mean and standard deviation. We observe that the standard deviations are relatively small, indicating that the randomness caused by dropout does not have a noticeable impact on the system.

**Impact of test set size.** We assume access to an unlabeled test set, so that a dataset-level BoS score can be computed. In the real world, we might access limited test samples, so we evaluate our system under varying test set sizes. Results are provided in Fig. 4(c). Our system gives relatively low RMSE ($\sim$2.54%) when the number of test images is more than 50.

**Hyperparameters selection**: dropout position $p \in \{[0], [1], [2], ..., [0, 1, 2, 3]\}$ and dropout rate $\epsilon \in [0, 1]$. For the former, there are 15 possible permutations for inserting dropout layer in 4 backbone stages. "[0, 1]" means adding a dropout after stage 0 and stage 1 of the backbone respectively, and so on. We search the optimal configuration on the training meta set in a greedy manner, where the interval for $\epsilon$ is 0.05 and permutations for dropout position are evaluated one by one. *We use the configuration which results in the highest determination coefficient $R^2$ between mAP and BoS score during training*. Noting that hyperparameters are selected in training (see Section 4.1 for training

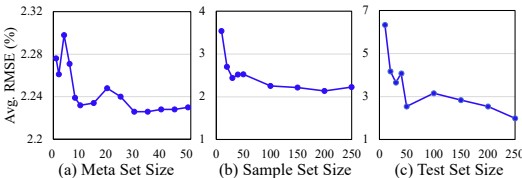

Figure 4: Impact of (a) meta-set size, (b) sample set size and (c) test set size on the performance of mAP estimators. RMSE (%) is reported. Generally speaking a larger meta-set and larger sample sets are beneficial for regression model training. Our system gives relatively low RMSE when the number of test images is more than 50.

meta-set details). Only for the purpose visualization convenience, we show how dropout rate $\epsilon$ and position $p$ affect the training coefficient of determination in Fig. 3(a) and (b), respectively. When visualizing the impact of one hyperparameter, we fix the other at optimal value found in training. For the RetinaNet+R50 vehicle detector, the best dropout rate $\epsilon$ is 0.15, and the best dropout position setting is adding a dropout layer after Stage 1 and Stage 2 ($p = [1, 2]$) of the backbone, respectively.

**Impact of meta-set size and sample set size.** The meta-set size is defined by the number of sample sets it contains, and the size of sample set is defined by the number of images it has. We test various meta-set and sample set sizes and report the results in Fig. 4(a) and (b), respectively. We observe that a larger meta-set size or sample set size leads to improved system performance measured by RMSE. Our results are consistent with (Deng & Zheng, 2021).

### 5.4 STRONG CORRELATION

One of our key contributions is the discovery of this strong relationship between mAP and BoS score. To study the relationship between mAP and BoS score on various test sets, we first train a detector, where we adopt RetinaNet+R50 (Lin et al., 2017) trained on the COCO (Lin et al., 2014).

Second, we construct a meta set. The meta-set is a collection of sample sets. To construct the meta-set, we collect various real-world datasets that all contain the vehicle categories as seed sets, including BDD (Yu et al., 2020), Cityscapes (Cordts et al., 2016), Kitti (Geiger et al., 2013) and so on (all datasets are listed in Fig. 2). To increase the number of datasets, we follow prior work (Deng & Zheng, 2021) to synthesize sample sets using various geometric and photometric transforms. Specifically, we sample 250 images randomly for each seed set, and randomly select three transformations from {Sharpness, Equalize, ColorTemperature, Solarize, Autocontrast, Brightness, Rotate}. Later, we apply these three transformations with random magnitudes on the selected images. This practice generates 50 sample sets from each source, and the sample sets inherit labels from the seeds.

Then, we simply compute the mAP scores and BoS scores (Eq. 4) of the detector on these test sets. Finally, we plot these mAP scores against the BoS scores in Fig. 2. We clearly observe from Fig. 2(c) that BoS score has a strong correlation with mAP, evidenced by the high $R^2$ score and Spearman's Rank Correlation $\rho$. It means that, for a given detector, its detection accuracy would be high in environments where the predicted bounding boxes are resistant to feature dropout; Similarly,

the mAP score would be low in test sets where its bounding box positions are sensitive to model perturbation. If we think about this from the perspective of test difficulty for a given detector, results suggest that the test environment is more difficult if detector outputs are less robust model perturbation, and vice versa.

We then compare the correlation produced by BoS score with prediction score (Hendrycks & Gimpel, 2016) and average thresholded confidence (Garg et al., 2022). We observe that the correlation strength of our method is much stronger than the competing ones. Apart from the study in Fig. 2, more correlation results, including other types of detectors, other detection metrics like $mAP_{50}$, and other canonical detection tasks like pedestrian detection, are provided in the supplementary materials, where our method gives consistently strong correlations.

## 5.5 DISCUSSIONS AND LIMITATIONS

**Is BoS score a kind of confidence/uncertainty measurement in object detection?** It characterizes confidence on the dataset level, but might not operate on the level of individual bounding boxes/images. In our preliminary experiment, we tried to correlate BoS score with accuracy on individual images and even bounding boxes, but the correlation was a little weak. We find BoS score is correlated with accuracy when the number of images considered is statistically sufficient.

**Using MC dropout, can we measure the change of bounding box confidence?** Yes. By bipartite matching, it is technically possible to compute the confidence difference between corresponding bounding boxes. But we find that this confidence stability score (CS score) does not correlate well with accuracy. Please refer to the bottom two pillars in Fig. 3(c) for this comparison.

**Can BoS score be a loss function for object detection during training?** No. There are two reasons. (1) the BoS-mAP correlation might not hold when a model just starts training, whose performance on a test set is extremely poor. (2) BoS score is not differentiable, since its calculation involves the perturbed detection results which are constantly changing during training. However, we might improve detector performance from the insights of our paper. For example, we can apply a resistance module like MC-DropBlock (Deepshikha et al., 2021) to improve detector generalization.

**Can other self-supervised signals exhibit a good correlation with mAP?** This work uses the pretext metric, box stability score, for label-free model evaluation, because it well correlates with mAP. There exist other existing pretext tasks for unsupervised object detection pre-training (Xiao et al., 2021; Dai et al., 2021; Bar et al., 2022). It is interesting to study in the future whether these methods also relate to mAP in target environments.

**Limitations on the working scope.** Strong correlation is observed when the detector has reasonably good performance (e.g., mAP > 15%) on the sample sets. If a test set is extremely difficult on which mAP could be as low as 1-2%, the correlation might not hold. The reason is when mAP is extremely low, there are too many missed detections. Without bounding boxes, BoS score cannot be computed. In fact, a similar observation is made in image classification: existing confidence-based indicators (Saito et al., 2019; Deng & Zheng, 2021) are less correlated with accuracy where classification accuracy is very low. Moreover, we might encounter images from unseen classes in the open world (Ji et al., 2023). In addition, the effectiveness of the BoS score in predicting the mAP for multi-class detectors requires further investigation. This requires *a large scale collection* of object detection datasets across *diverse appearances* and the *same label space*. However, real-world datasets meeting these criteria are scarce, which limits us from exploring more findings. Part of our future work is to collect such datasets like Sun et al. (2023) to share with the community.

## 6 CONCLUSION

In this paper, we report a very interesting finding: given a trained detector, its bounding box stability when feature maps undergo dropout positively correlates with its accuracy, measured on various test sets. The stability is measured by BoS score, computed as the intersection over union between corresponding bounding boxes found by bipartite matching. Because computing the BoS score does not require test labels, this finding allows us to predict detection mAP on unlabeled test sets. We perform extensive experiments to verify this capability and show that the mAP evaluator based on BoS score yields very competitive results.

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

## A  EXPERIMENTAL SETUP

### A.1  MODELS TO BE EVALUATED

We consider both one-stage and two-stage detection models with different backbones, including RetinaNet+R50 (Lin et al., 2017; He et al., 2016), RetinaNet+Swin (Liu et al., 2021), Faster-RCNN+R50 (Ren et al., 2015) and FasterRCNN+Swin. We follow common practices (Liu et al., 2021) to initialize the backbone with pre-trained classification weights, and train models using a $3\times$ (36 epochs) schedule by default. The models are typically trained with stochastic gradient descent (SGD) optimizer with a learning rate of $10^{-3}$. Weight decay of $10^{-4}$ and momentum of 0.9 are used. We use synchronized SGD over 4 GPUs with a total of 8 images per minibatch (2 images per GPU). During the training process, we resize the input images such that the shortest side is at most 800 pixels while the longest is at most 1333 (Chen et al., 2019). We use horizontal image flipping as the only form of data augmentation. During the testing process, we resize the input images to a fixed size of 800×1333.

### A.2  DATASETS

The datasets we use are publicly available and we have double-checked their license. Their open-source is listed as follows.
For vehicle detection, we use 10 datasets, including
**COCO** (Lin et al., 2014): `https://cocodataset.org/#home`;
**BDD** (Yu et al., 2020): `https://bdd-data.berkeley.edu/`;
**Cityscapes** (Cordts et al., 2015): `https://www.cityscapes-dataset.com/`;
**DETRAC** (Wen et al., 2020): `https://detrac-db.rit.albany.edu/`;
**Exdark** (Loh & Chan, 2019): `https://github.com/cs-chan/Exclusively-Dark-Image-Dataset`;
**Kitti** (Geiger et al., 2013): `https://www.cvlibs.net/datasets/kitti/`;
**Self-driving** (Kaggle, 2020a): `https://www.kaggle.com/datasets/owaiskhan9654/car-person-v2-roboflow`;

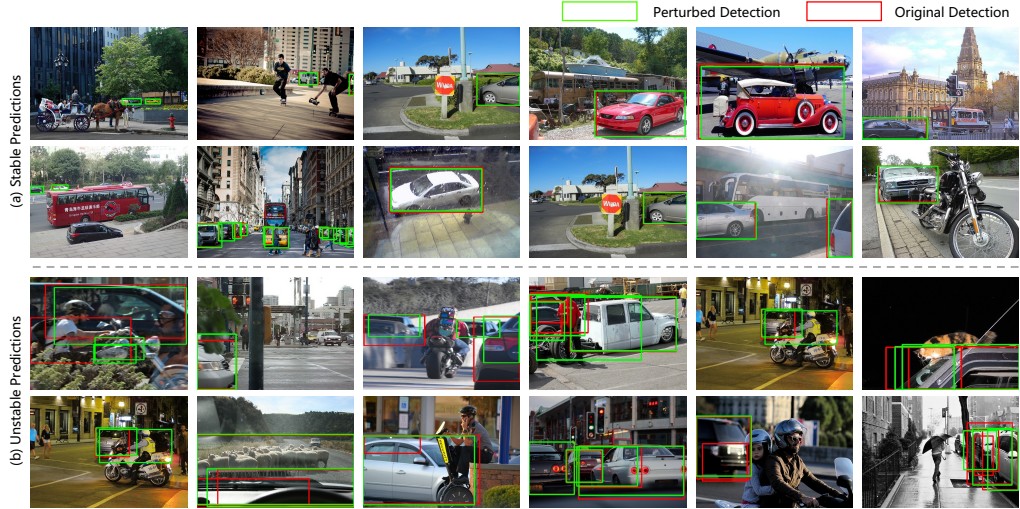

Figure 5: Visual examples illustrating the correlation between bounding box stability and detection accuracy. (a) Bounding boxes change significantly upon MC dropout, where we observe complex scenarios and incorrect detection results. (b) Bounding boxes remain at similar locations after dropout, and relatively correct detection results are observed.

**Udacity** (Kaggle, 2021): https://www.kaggle.com/datasets/alincijov/self-driving-cars;
**Roboflow** (Kaggle, 2022): https://www.kaggle.com/datasets/sshikamaru/udacity-self-driving-car-dataset;
**Traffic** (Kaggle, 2020b): https://www.kaggle.com/datasets/saumyapatel/traffic-vehicles-object-detection.
For pedestrian detection, we use 9 datasets, including
**COCO** (Lin et al., 2014): https://cocodataset.org/#home;
**Caltech** (Griffin et al., 2007): https://data.caltech.edu/records/f6rph-90m20;
**Crowdhuamn** (Shao et al., 2018): https://www.crowdhuman.org/;
**Cityscapes** (Cordts et al., 2015): https://www.cityscapes-dataset.com/;
**Self-driving** (Kaggle, 2020a): https://www.kaggle.com/datasets/alincijov/self-driving-cars;
**Exdark** (Loh & Chan, 2019) : https://github.com/cs-chan/Exclusively-Dark-Image-Dataset;
**EuroCity** (Braun et al., 2019): https://eurocity-dataset.tudelft.nl/;
**Kitti** (Geiger et al., 2013): https://www.cvlibs.net/datasets/kitti/;
**CityPersons** (Zhang et al., 2017): https://www.cityscapes-dataset.com/downloads/.

### A.3 COMPUTATIONAL RESOURCES

We conduct detector training experiments on four A100, and detector testing experiments on one A100, with PyTorch 1.9.0 and CUDA 11.1. The CPU is Intel(R) Xeon(R) Gold 6248R 29-Core Processor.

### A.4 HYPER-PARAMETER EXPLORATION FOR BASELINE METHODS.

We empirically find that different measurements have their own optimal thresholds. According to the effect of thresholds in terms of correlation strength ($R^2$) in Fig. 6, we use thresholds of 0.4, 0.95, and 0.3 for ATC (Garg et al., 2022),

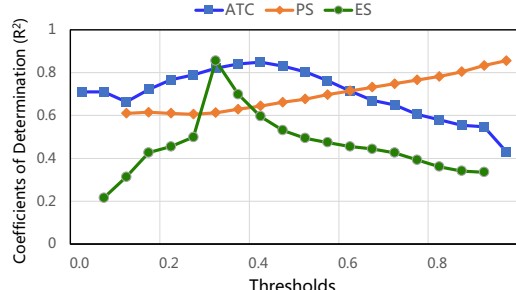

Figure 6: Effect of thresholds for all baseline methods. We search the optimal threshold on training sample sets using RetinaNet+R50. We report the correlation results (coefficients of determination $R^2$) using various thresholds. We show that threshold = 0.4, 0.95, and 0.3 helps for ATC, PS, and ES, respectively.

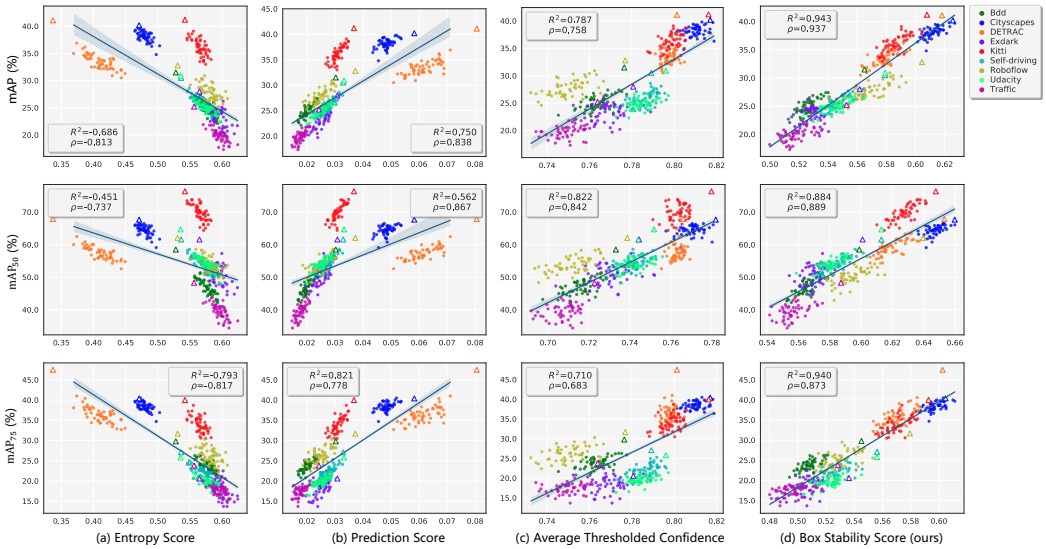

Figure 7: Correlation between different measurements and detection metrics for vehicle detection. "△" of different colors represents nine real-world datasets, such as BDD (Yu et al., 2020) and Cityscapes (Cordts et al., 2016), which are used as seeds to generate sample sets by various transformations. Each point "•" of different colors represents a sample set generated from different seed sets. Three detection metrics including mAP, $mAP_{50}$, $mAP_{75}$, which are obtained by RetinaNet (Lin et al., 2017) trained on the COCO training set. Four measurements including three existing methods (Saito et al., 2019; Hendrycks & Gimpel, 2016; Garg et al., 2022) and our box stability score are used in (a), (b), (c), and (d). The trend lines are computed by linear regression, from which we can observe a relatively strong linear relationship between box stability score and mAP.

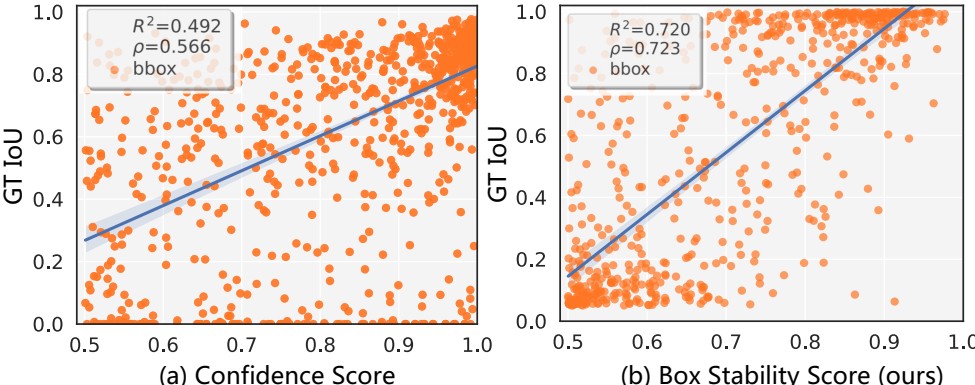

Figure 8: (a) The correlation between the IoU of bounding boxes with the matched ground truth and the confidence score. (b) The correlation between the IoU of bounding boxes with the matched ground truth and box stability score. Considering detected bounding boxes having a confidence/box stability score ($> 0.5$), the coefficients of determination $R^2$ are (a) 0.492, and (b) 0.720. Each point represents a predicted box generated by RetinaNet+R50 run on COCO validation without the detector feature perturbed.

PS (Hendrycks & Gimpel, 2016), and ES (Saito et al., 2019), respectively. We have two observations. First, ATC and ES tend to choose centering thresholds as their optimal thresholds, while ES is particularly sensitive to the setting of thresholds. Second, when using extremely high threshold temperature values, PS achieves its own strongest correlation with mAP.

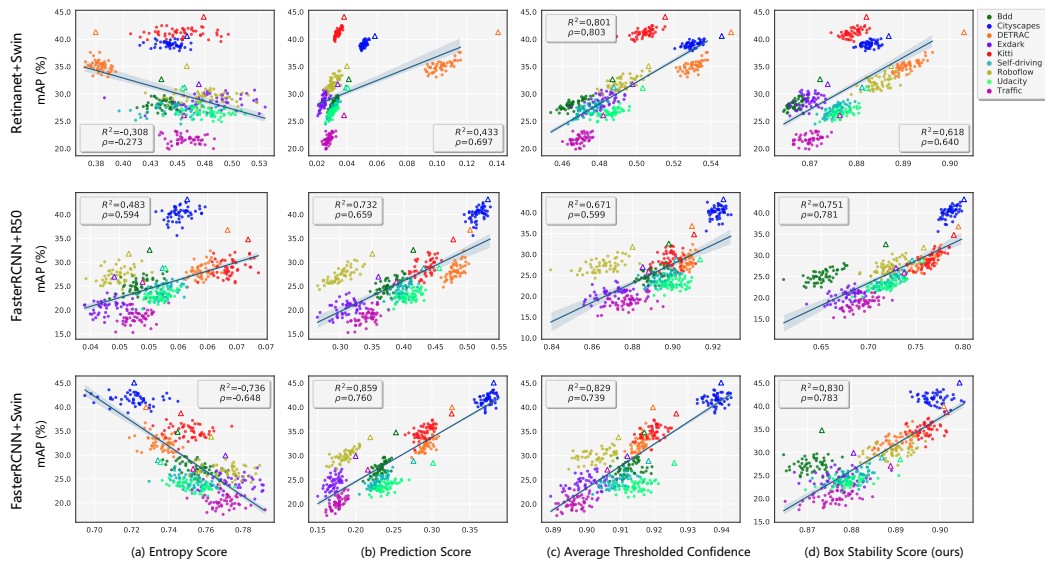

Figure 9: Correlation between different measurements and mAP on different structures for vehicle detection. "△" of different colors represents nine real-world datasets, such as BDD (Yu et al., 2020) and Cityscapes (Cordts et al., 2016), which are used as seeds to generate sample sets by various transformations. Each point "•" of different colors represents a sample set generated from different seed sets. The mAP of each raw is obtained by three detector structures trained on the COCO training set, including RetinaNet+Swin, FasterRCNN+R50, and FasterRCNN+Swin, respectively. Four measurements including three existing methods (Saito et al., 2019; Hendrycks & Gimpel, 2016; Garg et al., 2022) and our box stability score are used in (a), (b), (c), and (d). The trend lines are computed by linear regression, from which we can observe a relatively strong linear relationship between box stability score and mAP.

## B    DISCUSSION

**Is BoS score a kind of confidence/uncertainty measurement in object detection?** It characterizes confidence on the dataset level, but might not operate on the level of individual bounding boxes/images. As shown in Fig. 8(a), we tried to correlate box stability score with ground truth IoU (GT IoU) on individual bounding boxes. Compared to confidence score (coefficients of determination $R^2 > 0.49$), BS achieves a stronger correlation (coefficients of determination $R^2 > 0.72$) with GT IoU, but the correlation was still weak. We find BoS score is correlated with accuracy when the number of images considered is statistically sufficient (coefficients of determination $R^2 > 0.93$). In Fig. 5(a)(b), we give some visual examples of where BoS score can reflect whether the dataset is hard for the detector. Under feature perturbation, bounding boxes remaining stable (with high BoS score) would indicate a relatively good prediction, while bounding boxes changing significantly (with low BoS score) may suggest poor predictions.

## C    MORE CORRELATION RESULTS

### C.1    CORRELATION RESULTS FOR OTHER METRICS

In Fig. 7, we additionally show the correlation between different measurements and different detection metrics. We observe that BoS score also has a strong correlation with $mAP_{50}$ and $mAP_{75}$, yielding the other three measurements.

Figure 10: Correlation between different measurements and mAP for pedestrian detection. "△" of different colors represents nine real-world datasets, such as Caltech, Crowdhuman (Shao et al., 2018), Cityscapes (Cordts et al., 2015), and Self-driving (Kaggle, 2020a), which are used as seeds to generate sample sets by various transformations. Each point "●" of different colors represents a sample set generated from different seed sets. mAP is obtained by RetinaNet (Lin et al., 2017) trained on the COCO training set. Four measurements including three existing methods (Saito et al., 2019; Hendrycks & Gimpel, 2016; Garg et al., 2022) and our box stability score are used in (a), (b), (c), and (d). The trend lines are computed by linear regression, from which we can observe a relatively strong linear relationship between box stability score and mAP.

## C.2 CORRELATION RESULTS ON DIFFERENT DETECTORS

In this section, we additionally report the correlation results using other detectors: RetinaNet+Swin, FasterRCNN+R50, and FasterRCNN+Swin. As shown in Fig. 9, there is a very strong correlation between BoS score and mAP on FasterRCNN+Swin ($R^2 = 0.830$ and $\rho = 0.783$). But on RetinaNet+Swin and FasterRCNN+R50, the correlation is somehow weak, and we will try to improve on this in future work.

## C.3 CORRELATION RESULTS ON PEDESTRIAN DETECTION

In this section, we additionally report the correlation results on Pedestrian Detection. As shown in Fig. 10, the correlation between BoS score and mAP is significantly strong than the other methods when the mAP is greater than 15%. When mAP is less than 15%, the ATC method is slightly better among these four measures, but the sample sets are clustered together and hard to distinguish from each other.

