# OpenReview forum: "Bounding Box Stability against Feature Dropout Reflects Detector Generalization across Environments"
_ICLR.cc/2024/Conference — ICLR 2024 spotlight_

### Official Review · Reviewer_xRLp · 2023-10-26

**Soundness:** 2 fair
**Presentation:** 4 excellent
**Contribution:** 2 fair
**Rating:** 6
**Confidence:** 4

**Summary:**

This paper propose a box stability score (BoS score) for unsupervised detector evaluation on the real-world test sets. It has been well proved that BoS has a strong, positive correlation with detection accuracy measured by mAP.  BoS is the first metric proposed for unsupervised evaluation of object detection. Experiments demonstrate that it performs well with different network backbones and outperforms confidences based measurements.

**Strengths:**

1. This paper is well written and organized. The idea is clear and easy to follow.
2. Novelty: BoS is the first metric proposed for unsupervised evaluation of object detection.
3. Correlation between BoS and mAP is well demonstrated.
4. Good and consistent performance of BoS compared with exsiting confidence based metrics across meta-datasets, backbones, and iou thresholds.

**Weaknesses:**

1. My main concern on the proposed method is how it can be applied in the real world object detection and what it benefits. The proposed method has requirements on the detector's performance (mAP>15%) on target data. However, in real-world scenarios, we cannot be certain if the detector performs well enough to meet this requirement. We don't know how reliable is the BoS score. Can BoS be used to  improve the detection results?

2. Another issue is that the evaluation of BoS is primarily focused on single-class scenarios, and it does not perform as well in multi-class settings. In real-world applications, we often encounter multiple bounding boxes detected across different classes. Consequently, the accuracy of matching between the original prediction and the prediction after feature perturbation may be compromised. It remains uncertain whether the performance of BoS will degrade when dealing with multi-class data.

3. Additionally, the length of this paper exceeds the 9-page limit.

**Questions:**

1. In Table 3, under the multi-class detection setting on Cityscapes and Self-driving datasets, the BoS performs much worse than other methods on car. While under single class setting, the RMSE of BoS of car on Cityscapes is 0.89, which is much better than others.There is no explaination on this point. Does the proposed method unstable in multi-class setting?

2. In each table, does the number besides the dataset or class name reports the real mAP? For example, COCO (34.2) in Table 1. If so, in Table 2, on Kitti and CityPersons datasets, the mAP is 13.3 and 11.8 respectively, which are lower than the requirement (mAP > 15%).

---

> ### Author Response · Authors · 2023-11-21
> **Rebuttal from Authors of Paper4908 to Reviewer xRLp**
>
> ###
>
> > **Q1.** Limited operating range.
>
> **A1.** Thanks. As mentioned in the fifth paragraph of Sec. 5.5, our method works very well when mAP is greater than 0.15. We believe this could cover sufficiently many scenarios. As for extreme cases, we recognized the potential failure of our system: we cannot even compute BoS score if there are too many missed detections. That said, existing methods also face the same problem. As we make an early attempt at accuracy prediction in object detection, we believe future endeavors can be made to address this problem. In the revised version, we also discuss another two limitations on the working scope of BoS score in the fifth paragraph of Sec. 5.5. Please refer to our reply to Q2 of Reviewer h1Z9.
>
> > **Q2**. Can BoS be used to improve the detection results?
>
> **A2.** Great question. We have discussed this question in the third paragraph of Sec. 5.5 of our paper. Our answer is no. There are two reasons. (1) The BoS-mAP correlation might not hold when a model just starts training, whose performance on a test set is extremely poor. (2) BoS score is not differentiable, since its calculation involves the perturbed detection results which are constantly changing during training. However, we might improve detector performance from the insights of our paper. For example, we can apply a resistance module (like MC-DropBlock [1]) to improve detector generalization ability.
>
> ```
> [1] Deepshikha, Kumari, et al. "Monte Carlo dropblock for modelling uncertainty in object detection." arXiv preprint arXiv:2108.03614 (2021).
> ```
>
> > **Q3.** Performance in generic object detection with multiple classes.
>
> **A3.** As described in Sec. 5.2, BoS score can be easily extended to multi-class detectors theoretically because of its class-wise nature. To verify this, we should collect *a large scale* object detection datasets that have *diverse appearances* and the *same label space*. To answer this question, we implemented BoS on the 5 datasets we can find that contain pedestrians and vehicles simultaneously. The results in Table 3, where BoS is shown to be overall competitive compared with existing methods. In another aspect, we would like to emphasize there are very few real-world object detection datasets that satisfy the requirements of the *same label space*, so we primarily focused on single-class scenarios. We clarify this reason in Sec. 5.5 of our revised paper.
>
> > **Q4.** Page Limit.
>
> **A4.** Thanks. We have corrected it in the revised paper.
>
> > **Q5.** Is the proposed method unstable in multi-class setting?
>
> **A5.** When confronted with multi-category mAP predictions, the features between different categories may interfere with each other. Thus, it is expected that the performance on single-category datasets is better than multi-category for the proposed approach. Existing methods also face the same problem. For example, In Table 3, under the multi-class detection setting on the Kitti dataset, the method of prediction score (0.99% -> 2.26%), entropy score (2.23% -> 4.71%), average confidence (1.69% -> 3.2%) and Fr´echet distance (2.98% -> 3.35%) all perform much worse than their performance on single-class pedestrian detection setting in Table 2. We appreciate the reviewer’s comment and have added an explanation on this point to the paper.
>
> > **Q6.** Does the number beside the dataset or class name report the real mAP? Compared to Table 1, the mAP on Kitti and CityPersons datasets in Table 2 is low.
>
> **A6.** Yes, the number beside the dataset or class name reports the real mAP. The condition of mAP  > 15% is not a strict mandatory condition for mAP estimation. But when mAP < 15%, the performance of BoS including other existing metrics might be worse than when mAP > 15%. To compare the mAP estimation performance for pedestrian detection in Table 2, we train a detector on a CrowdHuman training set (mAP = 33.5%) and report its ground truth mAP (%) on different unseen test sets in the header (besides the dataset). Since the domain shift, the detector performance on Kitti and CityPersons datasets drops to 13.3% and 11.8%, respectively.

---

> > ### Comment · Reviewer_xRLp · 2023-11-21
> > **Thanks for the response.**
> >
> > Considering those limitations mentioned above, could the authors provide a clear explanation of the applications of BoS?

---

> ### Author Response · Authors · 2023-11-22
> **Rebuttal from Authors of Paper4908 to Reviewer xRLp**
>
> Many thanks for the further question. Application wise, BoS can be used to assess detection performance, where no test ground truths are given. By focusing on this 'last-mile' problem in model deployment, our system has the potential to identify scenarios where the detector deviates from its optimal states (in-distribution performance), and how much the deviation is. This will allow users to have a quick understanding of their system performance in the wild, which informs their decision making.
>
> Admittedly, as an early attempt, our system is still not perfect: low-end accuracy (mAP < 15%) and multi-class detectors are non-trivial to estimate, due to difficulty in estimating recall and finding class-shared datasets (Sec. 5.5). But, through the opportunity of sharing our findings, we believe joint efforts in the community will allow this problem to be solved in the end.

---

> > ### Comment · Reviewer_xRLp · 2023-11-22
> > **final rating**
> >
> > Thanks. I tried my best to understand where to use the BoS score, but I found: 1) Detection inherently need to handle multi-class scenarios. This is different from classification that one image only has one image-level lable. Considering the BoS is proposed for detection, I think this weakness will largely limit its application.
> > 2) In real deployment, a small validation set is always hold. BoS can hardly perform over mAP on the validation set. In wild, BoS may also need to tune the linear regressor for better mAP correlation. Moreover, I don't know if two well-trained detectors can have significant difference in BoS score, particularly considering the performance degradation in multi-class scenario.
> >
> > Therefore, I don't see much application potential for the current version of BoS score. But I expect a better version in the future, such as designed to be differetiable that can be directly used for unsupervised detector training or training with noisy data labels. Considering the work in this paper is inspiring and technically solid, I will maintain my final rating.

---

> > > ### Public Comment · ~Yang_Yang61 · 2023-11-23
> > > **Rebuttal from Authors of Paper4908 to Reviewer xRLp**
> > >
> > > Thank you very much for your reply. Your comments have been invaluable in enhancing the quality of our work. We have included additional discussions in the revised paper, ensuring a more informative presentation.

---

### Official Review · Reviewer_mbpe · 2023-10-29

**Soundness:** 3 good
**Presentation:** 2 fair
**Contribution:** 2 fair
**Rating:** 6
**Confidence:** 3

**Summary:**

This paper find that under feature map dropout, good detectors tend to output bounding boxes whose locations do not change much, while bounding boxes of poor detectors will undergo noticeable position changes. The proposed box stability score has a strong, positive correlation with detection accuracy.

**Strengths:**

+ This paper is clear and easy to read.

**Weaknesses:**

- I find it hard to understand the experiments. In Table 1, the authors say "we report its ground truth mAP (%) on different unseen test sets in the header, the estimated mAP (%) in the main table". However, the numbers for estimated mAP reported in the main table are very low. So what do the numbers in Table 1 represent for? Same for Table 2, Table 3 and Table 4. What are the best numbers in each column?

**Questions:**

- Why there is a big gap between dropout rate=0.1 and 0.15 in Figure 3 (b)?
- Why there is a peak at around 5 in Figure 4 (a)?

---

> ### Author Response · Authors · 2023-11-21
> **Rebuttal from Authors of Paper4908 to Reviewer mbpe**
>
> ###
>
> > **Q1.** Meaning of the numbers in Table 1, Table 2, Table 3, and Table 4.
>
> **A1**. Thanks for the reviewer's careful review. We will clarify it in the revised paper. In Table 1, we report ground truth mAP (%) on different unseen test sets in the header and the RMSE (%) in the main table. The lower the numbers reported in the main table are,  the worse the mAP estimation performance is. The best number in each column is the lowest one. The same for Table 2, Table 3, and Table 4.
>
> > **Q2.** Why there is a big gap between the dropout rate=0.1 and 0.15 in Figure 3 (b)?
>
> **A2.**  Hyperparameters of BoS Score like dropout rate are selected during training (see Sec. 4.1 for training meta-set details). The results of Figure 3 (b) are actually based on empirical experiments. On further analysis, we find that the gap between the dropout rate=0.1 and 0.15 in Figure 3 (b) might be caused by the Roboflow dataset. If we remove sample sets generated by the Roboflow dataset in the training meta set (See M8 in the table below), the big gap disappears and the trend curve has smoothed out. In Table 1, we also find that the Roboflow dataset is the most difficult dataset for BoS to predict mAP.
>
> |              | M1      | M2       | M3        | M4             | M5         | M6         | M7               | M8           | M9          | M10         |
> | ------------ | ------- | -------- | --------- | -------------- | ---------- | ---------- | ---------------- | ------------ | ----------- | ----------- |
> | Dropout Rate | w/o BDD | w/o COCO | w/o Kitti | w/o Cityscapes | w/o DETRAC | w/o Exdark | w/o Self-driving | w/o Roboflow | w/o Udacity | w/o Traffic |
> | 0.05         | 0.944   | 0.907    | 0.893     | 0.892          | 0.904      | 0.939      | 0.924            | 0.92         | 0.928       | 0.897       |
> | 0.1          | 0.921   | 0.877    | 0.844     | 0.843          | 0.85       | 0.894      | 0.896            | 0.935        | 0.897       | 0.851       |
> | 0.15         | 0.96    | 0.936    | 0.921     | 0.918          | 0.924      | 0.947      | 0.942            | 0.939        | 0.947       | 0.924       |
> | 0.2          | 0.951   | 0.938    | 0.916     | 0.911          | 0.918      | 0.935      | 0.94             | 0.936        | 0.944       | 0.918       |
> | 0.25         | 0.936   | 0.932    | 0.911     | 0.898          | 0.909      | 0.92       | 0.93             | 0.929        | 0.933       | 0.905       |
> | 0.3          | 0.911   | 0.924    | 0.891     | 0.871          | 0.886      | 0.894      | 0.916            | 0.911        | 0.916       | 0.885       |
>
> (The coefficients of determination $R^2$ between BoS and mAP are reported on different training meta-sets. M means "training meta-set" and the leave-one-out evaluation procedure is employed.)
>
> > **Q3.** Why there is a peak at around 5 in Figure 4 (a)?
>
> **A3.** As described in Sec. 4.2, from the strong correlation of BoS and mAP, we train a linear regression model as the mAP estimator on the constructed training meta-set to fit this relationship. As shown in the table below, the performance of the mAP estimator (measured by RMSE) can suffer from some fluctuations due to higher prediction variance when the meta set size is small. The peak at around 5 also means the performance of the mAP estimator fluctuates when the meta set size is too small. Nevertheless, we observe that as the meta set size is increased, both the RMSE and the variance in RMSE are reduced which leads to improved system robustness.
>
> | Meta Set Size             | 1     | 2     | 4     | 6     | 8     | 10    | 15    | 20    | 25    |
> | :------------------------ | ----- | ----- | ----- | ----- | ----- | ----- | ----- | ----- | ----- |
> | Avg. RMSE (%)             | 2.276 | 2.261 | 2.298 | 2.271 | 2.239 | 2.232 | 2.234 | 2.248 | 2.24  |
> | Variance of Avg. RMSE (%) | 1.110 | 0.930 | 0.050 | 0.110 | 0.040 | 0.020 | 0.160 | 0.002 | 0.002 |
>
> ###

---

> ### Comment · Reviewer_mbpe · 2023-11-21
> **Further Questions**
>
> Thank you for your response. I have raised my rating and still have some questions.
>
> 1. Why Roboflow is more difficult than others?
> 2. The performance looks nice in Table 1, but why it seems hard to perform well on pedestrian detection in Table 2?
> 3. Why BoS is combined with PS in Table 1, Table 2 and Table 3? What do you want to illustrate?
> 4. If I have understood correctly, you need to train the w0 and w1 for estimating BoS using training meta-set. Do you train a set of w0 and w1 for a specific task like vehicle detection using some samples from several datasets? So the method cannot be directly trained by various tasks at once and then tested on different tasks.
> 5. You use a pre-trained object detector without re-training for it?
> 6. Do different tasks result in different w0 and w1?
>
> Besides, I have a suggestion for the paper presentation. I do think you should make the font be bold for the best performance in each column in the tables for easier understanding.

---

> > ### Author Response · Authors · 2023-11-22
> > **Rebuttal from Authors of Paper4908 to Reviewer mbpe**
> >
> > > **Q4.** Why Roboflow is more difficult than others?
> >
> > **A4.** Great question. We cannot give a comprehensive explanation for now. A possible reason why Roboflow has low performance in mAP prediction is that Roboflow is somehow *different* from other datasets, making it deviate from the mAP regression line. Such difference may be reflected in many aspects, e.g., Roboflow has the lowest image aspect ratio (1.14) among all the datasets. Note that this *difference* may not have the same meaning as the *domain gap* that we often use. To better understand this, we need to collect more datasets with vehicles and analyze the patterns of failure cases. We believe this will be an important future work, which has been added to the Conclusion section.
> >
> > > **Q5.** The performance looks nice in Table 1, but why it seems hard to perform well on pedestrian detection in Table 2?
> >
> > **A5.** Again great question. First, we note that the overall error for mAP prediction in pedestrian detection and vehicle detection is 3.29% and 2.25%, respectively. We feel the gap is not significant. Second, for the 1.04% difference, a possible explanation is that pedestrian detection in inherently is a more difficult task than vehicle detection, which can be evidenced by their consistently lower mAP. A low mAP means some pedestrians are not detected, which poses difficulty for accuracy estimation methods including ours and existing ones.
> >
> > > **Q6.** Why BoS is combined with PS in Table 1, Table 2 and Table 3? What do you want to illustrate?
> >
> > **A6.** Thanks for the careful review. By doing so, we would like to study whether existing label-free evaluation methods (e.g., prediction score, or PS) are complementary to ours. From Table 1, Table 2 and Table 3, we find combining PS with BoS does not result in noticeable improvement. More results and discussions are provided in Figure 3 (c) and Sec. 5.2, respectively.
> >
> > > **Q7.** If I have understood correctly, you need to train the w0 and w1 for estimating BoS using training meta-set. Do you train a set of w0 and w1 for a specific task like vehicle detection using some samples from several datasets? So the method cannot be directly trained by various tasks at once and then tested on different tasks.
> >
> > **A7.** Thanks for the question.
> >
> > (1) You are correct that are need to train w0 and w1 for estimating mAP from BoS, using a training meta set.
> >
> > (2) Yes, for each type of object in the detection task, e.g., vehicle, we need to train a set of w0 and w1. These two parameters will be different for different objects. To optimize the two parameters, we use sample sets generated from possibly many seed datasets using image transformations. It is better if sample sets have more images.
> >
> > (3) Technically, it is possible to train, say, ten sets of w0 and w1, for a 10-way detection problem, and then apply the ten mAP estimators to predict mAP of detecting objects included in the 10 classes. It is not possible to predict detector accuracy of an object category outside the ten classes.
> >
> > Note, further, that training an mAP estimator for ten classes needs seed datasets that all have the ten classes, which our community does not currently have many. For example, if we consider the 20 classes in Pascal-VOC, only a few datasets, such as COCO and Object365, have these 20 classes. Such a limited number of domains hinders the training of comprehensive mAP estimators. This is why we only studied detectors with up to 2 classes in this work. We are doing further work along this line to introduce datasets of various domains and with the same classes.
> >
> > > **Q8.** You use a pre-trained object detector without re-training for it?
> >
> > **A8.** As described in the first paragraph of Sec. 5.4 and the caption of Table 1, the detector is trained on a COCO training set, and our target is to predict its mAP on different unseen test sets. In supplementary material, we provide more details in Sec. A.1 of “Models To Be Evaluated”. We follow common practices to initialize the backbone with pre-trained classification weights, and train models using a 3× (36 epochs) schedule by default.
> >
> > > **Q9.** Do different tasks result in different w0 and w1?
> >
> > **A9.** Yes, different tasks will result in different w0 and w1. Because we need to train an mAP estimator on the constructed training meta-set to fit this relationship for different tasks.
> >
> > > **Q10.** Suggestion for the paper presentation.
> >
> > **A10.** Great suggestion. We have made the font bold for the best performance in each column in the tables in the revised version.

---

> > > ### Comment · Reviewer_mbpe · 2023-11-22
> > > **Thanks for the response**
> > >
> > > The answers have addressed my questions.

---

> ### Comment · Reviewer_mbpe · 2023-11-22
> **Final Rating**
>
> **Strengths**
>
> * All the reviewers find that this work is the first to estimate the detector performance in an unlabeled test domain, and it discovers that good detectors tend to output bounding boxes whose locations do not change much under feature dropout.
> * All the reviewers also recognize that the proposed box stability score is better than other alternatives generally for vehicle detection and pedestrian detection
>
> **Weaknesses**
>
> * I agree with h1Z9 that the proposed box stability score cannot estimate the reliability of individual bounding boxes well but performs better when the whole dataset is given.
> * I agree with xRLp that the proposed box stability score is not impressive in multi-class scenarios, while the task of object detection is inherently multi-class.
> * In the authors' response to me, I find that the proposed box stability score does not have a good generalization ability. Each type of object needs to train a specific set of parameters. If we want to work on 10 classes, then we need to collect additional data from different domains to train the parameters for these 10 classes. This is unrealistic in object detection.
>
> Considering the task of assessing the detector is new and the performance on vehicle detection and pedestrain detection is generally better than other alternatives, I raised my rating to 6, marginally above the acceptance threshold, because I think this work is inspiring for future work. However, rejecting this paper is also fine for me due to its limitation in evaluating individual bounding boxes (h1Z9), multi-class scenarios (xRLp) and generalization ability to various object classes (me).

---

> ### Author Response · Authors · 2023-11-23
> **Rebuttal from Authors of Paper4908 to Reviewer mbpe**
>
> > **Q11.**  Limitation in evaluating individual bounding boxes.
>
> **A11.** We would like to clarify that results for evaluating individual bounding boxes are presented in Fig. 8 of the supplementary material. The coefficient of determination is $R^2>0.72$ between BoS and GT IoU, the latter is a tentative way of evaluating the quality of individual bounding boxes. This $R^2$ is much higher than the traditionally used confidence score method ($R^2>0.49$). For more details, please refer to our reply A1 to reviewer h1Z9.
>
> > **Q12.**  Limitation in multi-class scenarios
>
> **A12.**  We appreciate the reviewer’s comment aboult multi-class scenarios and have added explanations on this point to the paper ensuring a more informative presentation (please refer to our reply A3 to reviewer xRLp).
>
> > **Q13.**   I find that the proposed box stability score does not have a good generalization ability. Each type of object needs to train a specific set of parameters. If we want to work on 10 classes, then we need to collect additional data from different domains to train the parameters for these 10 classes. This is unrealistic in object detection.
>
> **A13.** Thanks for this comment. To address the lack of data, we use image transformations to generate various sample sets, following [2]. If we aim at more categories, we need to collect some datasets (*e.g*., 10 datasets) that have these categories and then use image transformations to generate more sample sets for regression training, which has reasonable feasibility. In future our focus will be collecting more datasets sharing more common categories and deepen our understanding of mAP prediction of various object classes and environments. We feel this will make useful contributions to the community.
>
> Thanks for the summary and rating. We really appreciate your thoughts and discussions during this period, which allow us to think deeper into our method.
>
> ```
> [2] Weijian Deng and Liang Zheng. Are labels always necessary for classifier accuracy evaluation? In Proceedings of the IEEE/CVF Conference on Computer Vision and Pattern Recognition (CVPR), pp. 15069–15078, June 2021.
> ```

---

### Official Review · Reviewer_h1Z9 · 2023-10-30

**Soundness:** 3 good
**Presentation:** 3 good
**Contribution:** 3 good
**Rating:** 8
**Confidence:** 3

**Summary:**

This paper provides a novel way to evaluate object detection systems without ground truth data, based on the stability of bounding boxes, and shows promising results in estimating detection accuracy. The authors propose a method for evaluating the quality of bounding boxes generated by object detection models and their correlation with detection accuracy. The paper introduces a metric named"box stability score" (BoS) and demonstrates a strong positive correlation between BoS scores and detection accuracy (mAP). The method is the first for AutoEval in object detection. Experiments on vehicle and pedestrian detection tasks demonstrate the effectiveness of the BoS score in estimating mAP and outperforming confidence-based measurements.

**Strengths:**

1. This paper proposes a metric, the box stability score (BoS), for object detection. The metric is validated that it has a strong positive correlation with the detection accuracy (mAP). The correlation provides an agent for evaluating detectors on real-world datasets lacking ground truth bounding boxes. It can better test the generalization performance of various detectors. It first applies the AutoEval algorithms for object detection.

2. The paper is well organized. The proposed method BoS is simple but effective for label-free detection evaluation. The discussions are sufficient to explain the insights of the paper.

3. The paper is well written. It's easy to follow the main contributions of the paper.

4. The experiments compare BoS with extended classification AutoEval methods. The comparison demonstrates that the BoS can better measure the detection tasks under label-free scenarios.

**Weaknesses:**

1. The BoS is correlated with the accuracy on the dataset level but failed on the individual level. It does not make sense. This part needs more explanation and provides more evidence.

2. A limitation section is needed for a comprehensive analysis.

**Questions:**

nil

---

> ### Author Response · Authors · 2023-11-21
> **Rebuttal from Authors of Paper4908 to Reviewer h1Z9**
>
> ###
>
> > **Q1.** The reason why the BoS is correlated with the accuracy on the dataset level but failed on the individual level.
>
> **A1.** Thanks for the question. For individual bounding boxes, as shown in Fig. 8 of our supplementary material, we tried to correlate box stability score with ground truth IoU (GT IoU) on individual bounding boxes. Compared to confidence score (coefficients of determination $R^2>0.49$ ), BoS achieves a stronger correlation (coefficients of determination $R^2>0.72$) with GT IoU. The result of "coefficients of determination $R^2>0.72$" indicates *BoS could predict detector performance on individual bounding boxes to a certain extent.* However, it is weaker when compared to the correlation on the dataset level (coefficients of determination $R^2>0.93$). This is because, at the dataset level, the impact of the randomness of BoS score is averaged out over many bbox samples, leading to a more stable and statistically significant measure of box stability. In contrast, for the level of individual bounding boxes, the box stability is measured on a single bbox sample and is thus more susceptible to the inherent randomness.
>
> > **Q2.** A limitation section is needed for a comprehensive analysis.
>
> **A2.** We agree with the reviewer that a limitation section is needed. In the original 'discussions' section, we have mentioned some limitations of BoS score about its performance on the individual level, indifferentiable character, and working scope. In the revised version, we renamed the 'discussions' section to the 'discussions and limitations' section for better understanding. Besides, we add two other limitations on the working scope of BoS score. (1) We might encounter images from unseen classes in the open world. To improve the mAP estimation performance under this scenario, a feasible solution is using out-of-distribution detection techniques to detect and reject such novel images. (2) The effectiveness of the BoS score in predicting the mAP for multi-class detectors requires further investigation. This requires *a large scale collection* of object detection datasets across *diverse appearances* and the *same label space*. However, real-world datasets meeting these criteria are scarce, which limits us from exploring more findings. Part of our future work is to collect such datasets to share with the community.

---

> > ### Comment · Reviewer_h1Z9 · 2023-11-23
> > **Response**
> >
> > Thanks for your response. The response has addressed my concerns. The paper is interesting and technically solid. I will keep my rating.

---

> > > ### Public Comment · ~Yang_Yang61 · 2023-11-23
> > > **Rebuttal from Authors of Paper4908 to Reviewer h1Z9**
> > >
> > > Thank you for reviewing our paper and the positive feedback. We are glad that the reviewer found the method interesting and technically solid.

---

### Official Review · Reviewer_19ZD · 2023-11-01

**Soundness:** 4 excellent
**Presentation:** 4 excellent
**Contribution:** 4 excellent
**Rating:** 10
**Confidence:** 4

**Summary:**

The authors discover and examine the phenomenon that - in an object detection setting - the stability of bounding box detection (in the presence of MC dropout variability) is strongly correlated to the ability of the detector to generalize to unseen datasets. Aside from a comprehensive set of experiments comparing the approach to various alternatives, the authors also spend time discussing what their insights are into the metric, as well as what it *cannot* do (a valuable insight into the metric),

**Strengths:**

- An important discovery since dropout is performed on the neural network and so is dataset-independent. The fact that it works makes perfect sense as the approach is effectively testing the detector for robustness to unexpected/untrained differences in internal weights, which  can occur with distinctively-different datasets (especially real data), and is something that dropout is essentially simulating.

- Results are comprehensive, covering multiple detection problems and comparing against several alternative algorithms.

- The authors also spend a couple of sections on summarising what their scoring metric really is, and what it could be used for (and importantly what it could NOT be used for ie: as a loss function). This kind of clear elucidation is invaluable.

**Weaknesses:**

- I really don't have any weaknesses to mention - I can fully see this becoming part of the standard toolkit for evaluating detectors

**Questions:**

- I'm struggling to come up with any question - the authors have done a very good job covering all the bases I could ask for

---

> ### Author Response · Authors · 2023-11-21
> **Rebuttal from Authors of Paper4908 to Reviewer 19ZD**
>
> Thank you very much for reviewing our work. We appreciate that the reviewer found our discovery important, results comprehensive, and elucidation clear.

---

> ### Comment · Reviewer_19ZD · 2023-11-30
> **Final summary and rating**
>
> After looking carefully through the other reviewer comments and interactions with the authors, I would summarise my final outlook as follows:
> - Regarding concerns from reviewers mbpe and xRLp over the training of a regression model from a 'meta-training-set' in order to evaluate a given detector's generalization to unseen datasets. The main issue appears to be the intricacy of the procedure needed to generate the estimate. However, I do not see this being a major issue: I find it highly analogous to N-fold cross-validation - sure it's a pain to split the data into N folds correctly stratified across classes, but if one wishes to properly evaluate a classifier then that's the gold standard. Similarly, in the case of BoS there is a fair amount of effort involved to build the meta-training set and fit the BoS regression model, but that's simply what needs to be done to take advantage of the BoS-to-mAP correlation that the authors have discovered. And it will spur new ideas on how to take advantage / evaluate in ways that are better and less troublesome.
>
> - On that note, I don't see the *core* of this paper to be the regressor/generalization evaluator at all, but rather the *discovery* by the authors of a correlation with BoS and mAP. This is a phenomenon independent of the author's use of it in the evaluator, and all reviewers agree both that the phenomenon appears to exist as the authors find, and that its use in a generalization evaluator would be extremely useful. My point is that the phenomenon itself needs independent verification by other parties to ensure it really is a thing/strong correleation and not some artifact of the author's test cases - hence the need to publish this. If it turns out to be valid/verifiable then this is *automatically* a landmark paper due to the sheer significance of such a strong correlation between a test-measurable metric (BoS) and a ground-truth evaluation (mAP). From there, there will of course be a flurry of papers that build better/simpler evaluators that what the authors put together, but that needs widespread knowledge of the phenomenon.
>
> - Thus one can argue over the details of how the authors have implemented their evaluator (ie: their use of the phenomenon), but the *phenomenon itself* is the thing that truly matters here, and everyone is in agreement that it if true it will have significant effects on the entire field (object detection) - whether the authors have made full use of the discovery or not is of secondary importance.
>
> - Hence I maintain my score (10) - the paper is technically solid in all areas and, most importantly, the authors have taken great pains to show the BoS-mAP correlation is probably real - that is what makes the paper immediately worthy of publication in my view since dissemination to, and independent verification by, the wider community is essential.

---

### Meta-Review · Area_Chair_ccXf · 2023-12-05

**Metareview:**

After review, rebuttal, and discussion all reviewers recommend acceptance. The AC agrees.
The paper is well written, novel, and experimentally well executed. The authors were able to answer all questions and concerns the reviewers raised.

**Justification For Why Not Higher Score:**

The paper is well executed and interesting, but may not appeal to a wider audience beyond object detection.

**Justification For Why Not Lower Score:**

The paper is interesting enough to a significant part of computer vision to be more than a poster.

---

### Decision · Program_Chairs · 2024-01-16

Accept (spotlight)